# Direct imaging of residual oxygen disorder in an infinite-layer nickelate superlattice via multislice ptychography

Chao Yang ✉, Hongguang Wang ✉, Roberto A. Ortiz, Kelvin Anggara, Eva Benckiser, Bernhard Keimer & Peter A. van Aken

Infinite-layer nickelates have garnered significant attention due to their potential for high-temperature superconductivity. Despite extensive research, the interplay between oxygen stoichiometry and electronic properties in infinite layer nickelates remains inadequately understood. In this study, we employ advanced electron microscopy techniques and theoretical modeling to directly visualize the distribution of residual oxygen within an $8NdNiO_2/2SrTiO_3$ superlattice, providing novel insights into its structural and electronic effects. Our multislice ptychography analysis reveals a disordered arrangement of apical oxygen atoms, even in regions with low residual oxygen occupancy, invisible in conventional projected images but discernible in depth-resolved phase contrast images. This disordered distribution suggests the formation of local domains with varying degrees of oxygenation, leading to significant structural distortions. Electron energy-loss spectroscopy reveals inhomogeneous hole doping in the infinite layer nickelates. Complementary density functional theory calculations show how residual oxygen and associated structural distortions—such as pyramidal and octahedral configurations—alter the electronic structure. Although superconductivity was not observed in the studied superlattice, our findings highlight the critical influence of residual oxygen in shaping electronic phases and suggest that precise control of oxygen stoichiometry is essential in infinite layer nickelates.

The investigation of complex oxide materials has garnered significant attention due to their fascinating electronic and physical properties. Among these, infinite-layer nickelates have emerged as a focal point in condensed matter physics, driven by their potential for high-temperature superconductivity achieved through hole doping[1]. Since their initial discovery, infinite-layer nickelates have been compared to the cuprate superconductors[2], with both systems featuring layered structures and unconventional superconductivity. A common strategy for inducing hole doping involves substituting cations at the A-site rare-earth elements (A: Nd, La, Pr), resulting in a $d^{8.8}$ electronic configuration associated with the superconducting phase in infinite-layer nickelate single films or superlattice[1,3–5]. Additionally, modifying the

stacking sequence in Ruddlesden-Popper phases has demonstrated some success in achieving superconductivity[6]. Despite these efforts, alternative approaches, including superlattice engineering[7,8] and anionic doping[9], have yet to yield superconducting phases, likely due to structural disorder.

Recent breakthroughs have expanded the superconducting phase diagram of nickelates, with reports of superconductivity in un-doped infinite-layer nickelates such as $NdNiO_2$[10], and $PrNiO_2$[11]. These findings challenge the prevailing understanding of superconductivity in nickelates and suggest a re-evaluation of the role of hole doping. Of particular interest is the hypothesis that residual oxygen may act as a hole-

Max Planck Institute for Solid State Research, Stuttgart, Germany. ✉e-mail: c.yang@fkf.mpg.de; hgwang@fkf.mpg.de

doping mechanism in these un-doped systems, mirroring the effects of more deliberate doping strategies.

Residual oxygen has emerged as a critical factor in the structural and electronic behavior of infinite-layer nickelates, attracting increasing attention. For instance, advanced scanning transmission electron microscopy (STEM) studies have revealed strong surface reconstructions in partially reduced nickelates[12], attributed to thickness-dependent oxygen deintercalation and its impact on surface polarity. Additionally, debates around the observed $3a_0$ charge density wave order in infinite-layer nickelates, supported by X-ray scattering and STEM measurements, have pointed to an ordered arrangement of residual apical oxygen caused by incomplete de-intercalation[13,14]. These findings underscore the importance of understanding the effects of residual oxygen in nickelates, particularly in the context of superconductivity.

Due to the low occupancy and uncertain distribution of residual oxygen in infinite-layer nickelates, conventional STEM techniques, such as annular bright field (ABF), integrated differential phase contrast (iDPC), and integrated center of mass (iCoM) imaging, are insufficient for providing depth-resolved information. For example, iCoM phase-contrast imaging of a NdNiO$_2$/SrTiO$_3$ super-lattice indicated the absence of apical oxygen, although subtle contrast at apical sites hinted at its potential presence[8]. Despite this, the infinite-layer structure was confirmed through quantitative analysis of out-of-plane lattice spacings[8].

In this work, we address these limitations by employing multislice ptychography, enabling direct imaging of the apical residual oxygen with occupancies below 12% in infinite-layer nickelates. By generating

sliced depth-resolved phase-contrast images, we reveal the presence of apical oxygen, which remains invisible in traditional projected iCoM and ptychography images. Our analysis indicates a random distribution of residual oxygen in the infinite nickelate layer. Furthermore, electron energy-loss spectroscopy (EELS) of the O $K$-edge spectra reveals an inhomogeneous pre-peak feature associated with hole doping. Combining these findings with density functional theory (DFT) calculations, we propose potential structural configurations induced by residual oxygen and explore their influence on the electronic structure.

## Results

Figure 1 presents the results of 4D-STEM image simulations using iCoM and multi-slice ptychography for a non-stoichiometric NdNiO$_{2+x}$ supercell with partial apical oxygen occupancy. These simulations highlight the differences in oxygen contrast attributable to residual apical oxygen. In practical experiments, focus conditions are typically assessed via high-angle annular dark-field (HAADF) imaging, which offers real-time feedback. However, HAADF imaging exhibits limited sensitivity to defocus due to its reliance on $Z$-contrast (atomic number contrast)[15–17]. Consequently, minor defocus is anticipated during 4D-STEM measurements. Fig. 1a illustrates the setup for 4D-STEM simulation of the infinite layer nickelates with residual oxygen at apical sites. Using diffraction patterns recorded by a virtual pixelated fast detector (Fig. 1a), the sample's atomic structure, including depth information, was reconstructed from simulated 4D datasets (Fig. 1b). The data reveal partially occupied apical oxygen sites and fully occupied basal oxygen sites. A structural model (Fig. 1c) was generated,

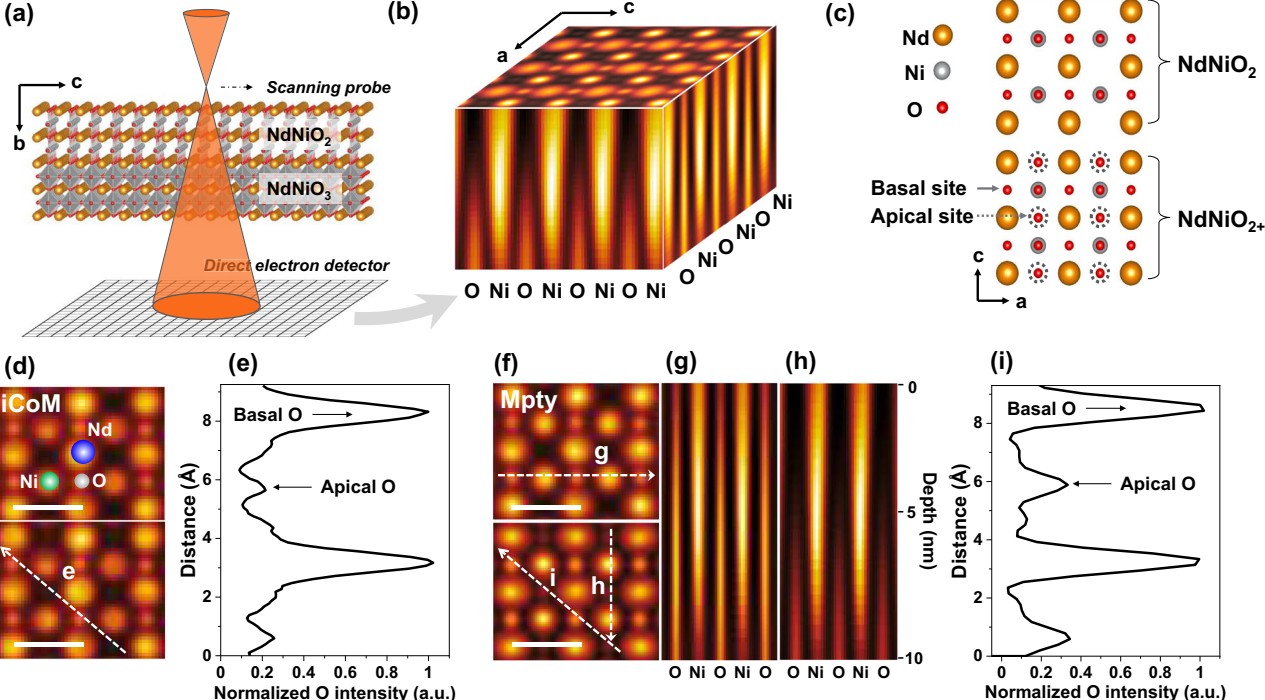

**Fig. 1 | Simulated multislice ptychography reconstruction of infinite layer nickelates with residual oxygen. a** Schematic representation of the 4D-STEM experimental setup, illustrating the use of an over-focused convergent electron probe scanning the sample in real space, while a Merlin direct detector captures the convergent diffraction patterns in Fourier space. **b** Reconstructed three-dimensional phase-contrast images of the NdNiO$_{2+x}$ sample, highlighting fully occupied basal oxygen sites and partially occupied apical oxygen sites. **c** Supercell structural model of the NdNiO$_{2+x}$ sample, showcasing the spatial arrangement of apical and basal oxygen atoms, used for integrated center of mass (iCoM) and multi-slice ptychography simulations. The dashed circles indicate the partially

occupied (35% occupancy in depth direction) apical oxygen. **d** Simulated iCoM image of the oxygen lattice. **e** Line profile of normalized oxygen signal intensity for basal and apical oxygen, extracted along the white dashed line **e** in **d**. **f** Phase-contrast image simulated using multi-slice ptychography (Mpty), capturing detailed depth information. The scale bars in image **d** and image **f** represent a length of 0.39 nm. **g** Basal oxygen depth profile along the white arrow **g** in **f**. **h** Apical oxygen depth profile along the white arrow **h** in **f**. **i** Line profile of normalized oxygen signal intensity for basal and apical oxygen, extracted along the white dashed line **i** in **f**. The scale bar in image **d** and image **f** represents a length of 0.39 nm.

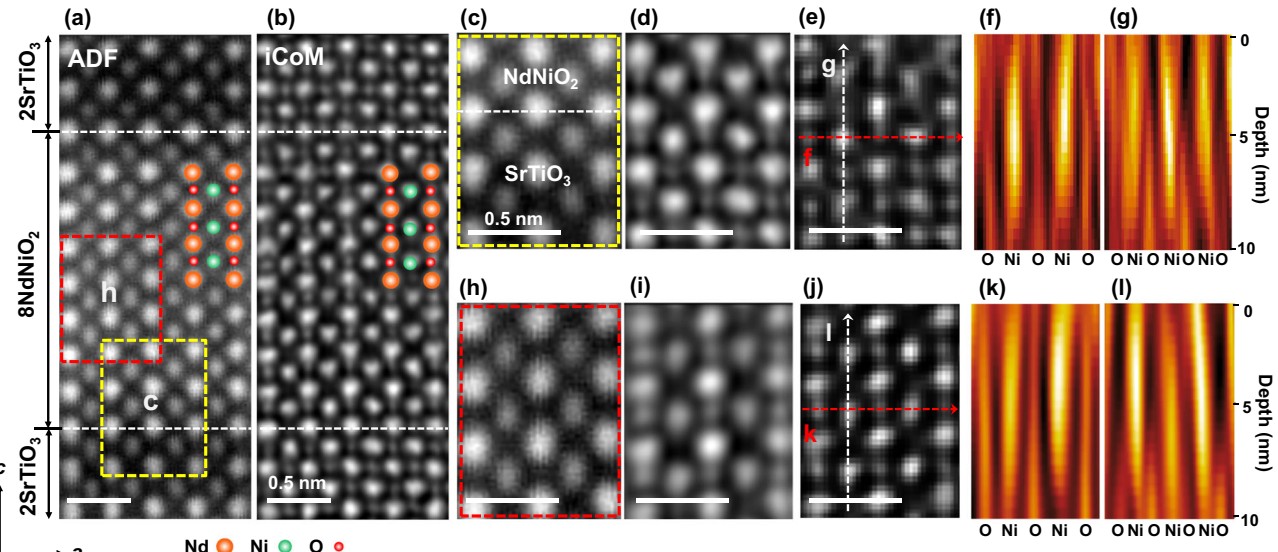

**Fig. 2 | Comparison of experimental visualization of the oxygen sub-lattice through reconstructed iCoM and multi-slice ptychography phase-contrast images, highlighting the direct imaging of residual apical oxygen along the *z*-direction in the 8NdNiO₂/2SrTiO₃ super-lattice film. a** Reconstructed annular dark field (ADF) image. **b** iCoM image derived from the 4D-STEM dataset. **c** Magnified ADF image of the SrTiO₃/NdNiO₂ interface region, outlined by the red dashed box in **a**. **d** Corresponding magnified iCoM image of the same region. **e** Phase-contrast image of the SrTiO₃/NdNiO₂ interface reconstructed using multi-slice ptychography, with accompanying depth profiles of **f** basal oxygen and **g** apical oxygen. **h** Magnified ADF image of the NdNiO₂ inner layer region, outlined by the yellow dashed box in **a**. **i** Corresponding magnified iCoM image of the NdNiO₂ inner layer. **j** Phase-contrast image of the NdNiO₂ inner layer region reconstructed using multi-slice ptychography, along with the depth profiles of **k** basal oxygen and **l** apical oxygen.

comprising an infinite-layer $NdNiO_2$ phase in the upper region and a $NdNiO_{2+x}$ perovskite phase with 35% apical oxygen occupancy in the lower region.

The simulated iCoM image reconstruction of this model (Fig. 1d) underestimates apical oxygen occupancy, showing approximately 23% oxygen concentration along the white dashed line (Fig. 1e). In contrast, the phase contrast image reconstructed via multi-slice ptychography (Fig. 1f) provides enhanced visibility of apical oxygen. In particular, depth profiles extracted from the basal and apical oxygen regions (Fig. 1g, h) reveal fully occupied basal oxygen sites, whereas apical oxygen is partially occupied in the lower regions of the sample. Signal intensity analysis (Fig. 1i) indicates that the reconstructed apical oxygen occupancy (~31%) aligns more closely with the structural model than with the iCoM. The discrepancy observed between the phase contrast and the structural model is closely related to the limitations of the contrast transfer function and its ability to handle dynamical scattering effects.

Additional simulations in the Supplementary Information (Fig. S1) compare various apical oxygen occupancies (4%, 12%, 23%, 35%, and 50%). It is notable that apical oxygen becomes effectively invisible in projected phase contrast images at concentrations below approximately 12%. Given a linear phase response between the contrast and oxygen concentration using electron multislice ptychography[18], it can be deduced that the concentration of the "invisible" apical oxygen is likely to be affected. Depth-resolved reconstructions (Figure S2) further illustrate the gradual emergence of apical oxygen contrast with increasing depth, emphasizing the importance of depth-sensitive imaging for low-occupancy configurations. Figure S3 shows another simulation with a different structural model of 10-nm-thick $NdNiO_{2+x}$ with 15% apical oxygen occupancy, where the residual oxygen is located solely in the central region. The reconstructed phase contrast images at various depths (Figure S3d) clearly reveal the distribution of residual oxygen, consistent with the structural model shown in Figure S3c. In addition, the depth of resolution achieved by multislice ptychography reconstruction has been demonstrated to be approximately 2.0 nm for oxygen dopants within a 15-nm-thick $PrScO_3$

structure[19]. Our simulations in Figure S3 yield a depth resolution comparable to previous studies, approximately 2.3 nm, allowing clear identification of the residual apical oxygen. Although the experimental resolution may be slightly degraded due to the signal noise, beam instabilities, and sample quality issues, it remains sufficient to identify residual oxygen in our samples.

Figure 2 showcases phase contrast reconstructions from 4D-STEM measurements of an 8NdNiO₂/2SrTiO₃ superlattice. Previous studies demonstrated the stabilization of infinite-layer structures in similar superlattices and capping-layer-stabilized films[8]. The ADF image in Fig. 2a confirms the high-quality sample, with the distinct NdNiO₂/SrTiO₃ interfaces. iCoM imaging (Fig. 2b) indicates the absence of apical oxygen in the NdNiO₂ inner layer, consistent with the infinite-layer structure. However, the underestimation of oxygen concentration particularly in defocused conditions by iCoM necessitated a detailed depth-resolved reconstruction.

Magnified ADF and iCoM images at the interface (Fig. 2c, d) reveal residual apical oxygen forming $NiO_5$ pyramidal units with minor polar distortion. Correspondingly, the strong interface polarity drives a displacement of the apical oxygen toward the Nd columns, leading to an elongated contrast of the atoms near the interface within the infinite-layer nickelate layer. This elongation contrast is not present in the SrTiO₃ layer. Multi-slice ptychography (Fig. 2e) enhances the apical oxygen contrast. Depth profiles (Fig. 2f, g) show fully occupied basal oxygen sites and a gradient in apical oxygen concentration, which decreases from the interface into the nickelate layer. Figure S4 shows the interface region, from which extracted slices of the phase-contrast images were taken at depths of 45 Å and 75 Å, respectively. Distinct residual apical oxygen, marked with white dashed circles in Figure S4e, is observed in the phase-contrast image at a depth of 75 Å, but not at a depth of 45 Å, as shown in Figure S4d. In the NdNiO₂ inner layer (Fig. 2h, i), apical oxygen is absent from the reconstructed iCoM image, a finding further confirmed by the uniform out-of-plane lattice spacing (~3.3 Å) in Figure S5, consistent with bulk NdNiO₂ and prior studies[8,20]. However, the missing apical oxygen in the projected phase contrast image reconstructed by multislice ptychography in Fig. 2j may also

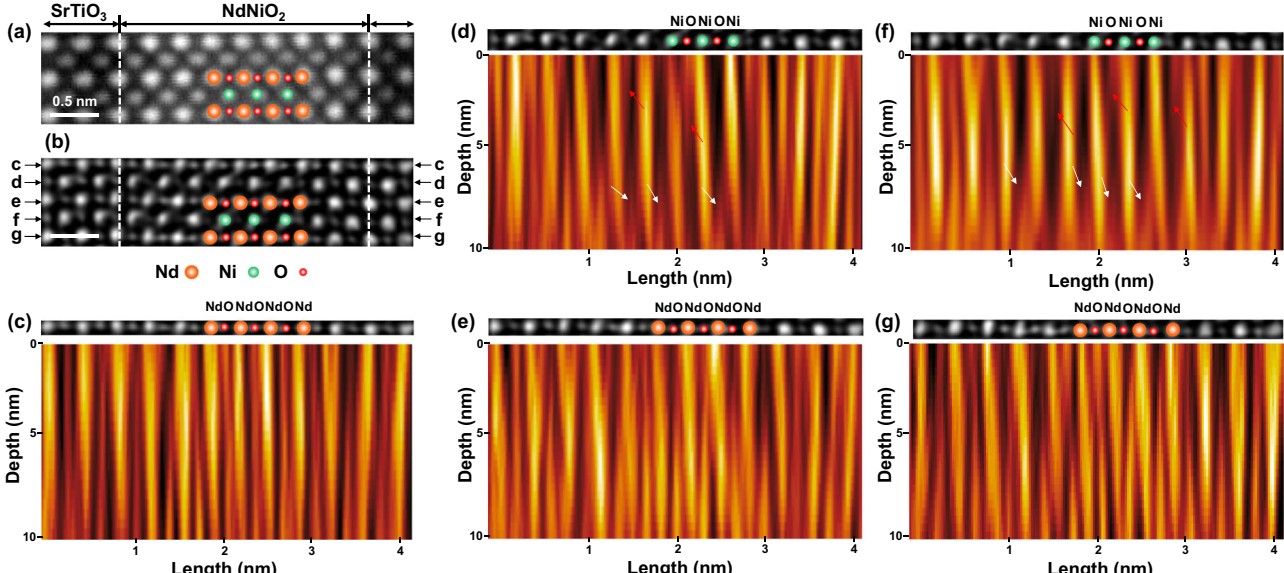

**Fig. 3 | Experimental visualization of oxygen disordering along the _z_-direction in the 8NdNiO₂/2SrTiO₃ superlattice film. a** Reconstructed annular dark field (ADF) image obtained from the 4D-STEM dataset, providing an overview of the atomic structure. **b** Projected phase-contrast image reconstructed using multi-slice ptychography, highlighting the oxygen sublattice with enhanced depth resolution. The depth profiles corresponding to multiple atomic layers are shown for **c** basal oxygen, **d** apical oxygen, **e** basal oxygen, **f** apical oxygen, and **g** basal oxygen as marked in **b**. The white and red arrows indicate the presence of residual apical oxygen along the _z_-direction, revealing the disordered nature of oxygen distribution within the super-lattice.

indicate a low occupancy of the apical oxygen, roughly less than twelve percent. The basal oxygen distribution from the depth profile image in Fig. 2k demonstrates no variation in depth direction. Notably, the depth-resolved phase contrast profiles (Fig. 2l) confirm the presence of residual apical oxygen hidden in the inner infinite nickelate layer in the superlattice. Multiple reconstructed slices (Figure S6) illustrate the variable presence of apical oxygen at different depths, as indicated by white dashed circles. Residual apical oxygen is present at 80 Å depth but absent at 55 Å. In addition, Figure S7 shows a different region of the NdNiO₂ inner layer. A residual apical oxygen atom, highlighted by a white dashed circle in Figure S7b, is observed in the projected phase-contrast image; this atom forms part of a pyramidal NiO₅ structure. The residual apical oxygen is clearly visible at depths of 35 Å and 50 Å, but its contrast noticeably decreases at 80 Å. These different configurations of residual oxygen in the infinite layer nickelate sample demonstrate the disordered nature of its distribution.

To obtain a comprehensive understanding of the depth and planar distribution of oxygen in the infinite-layer nickelate, we reconstructed the phase contrast images of basal and apical oxygen layers on a unit-cell-by-unit-cell basis using multi-slice ptychography, as illustrated in Fig. 3. The reconstructed ADF image in Fig. 3a provides a detailed visualization of the atomic structure at the NdNiO₂/SrTiO₃ interface, which was from the region marked with red a dashed box in Figure S8. Fig. 3b presents the corresponding projected phase contrast image, offering enhanced insight into the oxygen distribution.

Depth profile images were extracted from multiple layers, including regions labeled as basal oxygen (Fig. 3c, e, g) and apical oxygen (Fig. 3d, f). The depth profile in Fig. 3c highlights the NdO (or SrO) layer and basal oxygen distribution across the Nd (or Sr) lattice sites. Notably, basal oxygen appears to be approximately fully occupied, yet exhibits a discernible distortion along the depth direction. A comparable distortion pattern is observed in the basal oxygen profiles of Fig. 3e, g, reflecting consistent structural irregularities. The structural coupling between SrTiO₃ and NdNiO₂ induces a slight distortion in the oxygen sublattice of the SrTiO₃ layer along its depth. This distortion is absent in the oxygen depth profile of the SrTiO₃ substrate (Figure S9).

As illustrated in Fig. 3d, f, the apical oxygen sites in the SrTiO₃ layer are fully occupied, while the contrast gradually diminishes from the interface towards the inner NdNiO₂ layers. This residual apical oxygen in the infinite layer exhibits a disordered distribution, marked by the white and red arrows in Fig. 3d, f, indicating a disruption in the expected uniformity. Such disorder suggests that the nickelate layer, despite its predominant infinite-layer structure, incorporates a fraction of pyramidal and perovskite configurations due to residual oxygen incorporation. Furthermore, these residual oxygen atoms likely induce local atomic distortions, attributed to inhomogeneous electrostatic interactions. These distortions are reflected in the observed irregularities of basal oxygen distribution and are consistent with previously reported findings on the impact of non-stoichiometric oxygen content on structural stability and electronic properties of nickelate systems[8]. In addition, we conducted a statistical estimation of the oxygen occupancy based on the projected phase-contrast image in Figure S10. The oxygen sites in SrTiO₃ are considered fully occupied and can serve as the reference for 100% occupancy. Our estimation in Figure S10d suggests that residual oxygen occupancy is mostly in the range of 10 – 15%.

To investigate the impact of residual oxygen on the electronic structure of infinite-layer nickelates, we employed atomic-resolution STEM-EELS measurements complemented by DFT calculations, as shown in Fig. 4. The overview HAADF image in Fig. 4a shows the atomic structure of the NdNiO₂/SrTiO₃ superlattice after topotatical reduction. The magnified ADF image in Fig. 4b clearly delineates the NdNiO₂/SrTiO₃ interface, exhibiting distinct elemental contrast. Elemental maps extracted from Sr-$L_{2,3}$, Ti-$L_{2,3}$, Nd-$M_{4,5}$, and Ni-$L_{2,3}$ edges confirm the distinct distribution of elements with no detectable impurity phases or structural defects. There is one atomic layer of Ni-Ti intermixing at the interface, which leads to the increased residual oxygen observed in Fig. 2e.

The pre-edge region of O $K$-edge spectra, which are sensitive to the hybridization of O $2p$ and metal $3d$ states in nickelates, provide critical insights into the electronic structure[8,12,18,21–23]. Consistent with prior studies, the O $K$-edge in NdNiO₃ films exhibits a pronounced pre-peak at approximately 527.5 eV (Fig. 4c), indicative of a Ni $3d^8$L

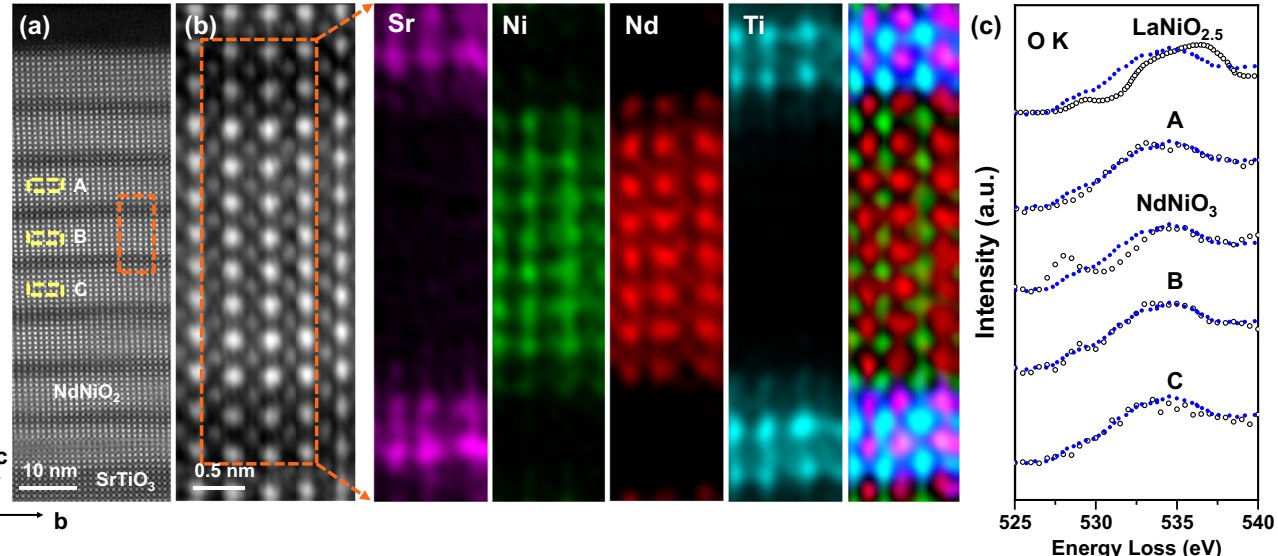

**Fig. 4 | Atomic-resolution determination of hole doping and the influence of residual oxygen on the electronic structure of the infinite layer nickelate superlattice using STEM-EELS. a** Overview and **b** magnified HAADF images showing the region used for EELS mapping. The orange dashed box in **a** shows the region cropped for **b**. Elemental EELS maps of Sr (purple), Ni (green), Nd (red), and Ti (cyan) are overlaid to reveal distinct elemental distributions at the atomic scale. **c** O $K$-edge spectra for $LaNiO_{2.5}$[42], $NdNiO_3$[23], and the regions labelled A, B, and C within the infinite inner layers of $NdNiO_2$ shown in **a**, highlighting changes in electronic structure associated with varying oxygen content. The blue reference spectra of O $K$ edge are from the hole-doped infinite layer $Nd_{0.775}Sr_{0.225}NiO_2$ film (PNAS, licensed under CC BY-NC-ND 4.0)[21].

electronic configuration[23]. However, in infinite-layer $NdNiO_2$ films, this pre-peak of O $K$ edge captured from the region A (Fig. 4a) is absent, which is a distinct feature of Ni $3d^9$ configuration of the reduced structure[8,12,18,21–23].

Residual oxygen plays a pivotal role in modifying electronic properties. Reports linking superconductivity in un-doped $LaNiO_2$ to residual oxygen highlight the significance of self-doping mechanisms mediated by oxygen defects[24]. Our analysis, based on phase-contrast imaging and simulations, estimates the residual apical oxygen concentration in our sample to be below 12%, which is close to the Ni $3d^{8.8}$ configuration in the $Nd_{0.8}Sr_{0.2}NiO_2$ film[1]. As shown in Fig. 4c, a spectral feature near 529 eV in the O $K$-edge of regions containing visible residual oxygen resembles features observed in hole-doped samples[21], implying a possible hole-doped electronic structure induced by the residual oxygen. Notably, the pre-peak is absent in the EELS spectra of superconducting $PrNiO_2$ samples, likely due to the inhomogeneous distribution of residual oxygen, which impacts measurement accuracy[11]. The observed disordered residual oxygen modifies the electronic structure irregularly, underscoring the complexity of oxygen's role in infinite-layer nickelates.

Given the uneven removal of oxygen ligands during the reduction process by annealing with $CaH_2$, a twin structure consisting of $a$-axis oriented and c-axis oriented infinite layers can form[25]. The $a$-axis oriented infinite layer structure was dominated in the $NdNiO_2$/$SrTiO_3$ superlattice, as shown by X-ray resonant reflectometry[26]. To further elucidate the possible effects of disordered residual oxygen in our sample, we performed DFT calculations to model various configurations. Figure S11a depicts an $NdNiO_{2.1}$ structure, including both $a$-axis and $c$-axis-oriented infinite-layer structures under the epitaxial strain imposed by the $SrTiO_3$ substrate, which were marked with b and d, respectively, as well as the domain interface marked with c. The $a$-axis-oriented structure shows significant distortion of the oxygen positions in the case of out-of-plane compressive strain from $c$-axis-oriented structure based on the experimental observation, whereas the $c$-axis-oriented structure remains largely undistorted. However, the $a$-axis-oriented structure is likely to be unstable under such high strain. This instability can be released by the formation of defects or cation

distortion, which could be a contributing factor to the observed absence of coexistence between $c$-axis and $a$-axis oriented structures. At the domain interface, pyramidal distortions arise, driven by additional oxygen incorporation, which introduces holes into the system.

In the $c$-axis structure (Figure S11c), it has been demonstrated that the low energy interactions are influenced by a complex contribution of Nd orbitals hybridized with Ni $d_{x^2-y^2}$ orbital, consistent with prior observations[27,28]. Conversely, in the $a$-axis-oriented structure, strong oxygen distortions can modify hybridization between Nd orbitals and the Ni $3d_{z^2}$ orbitals, as evidenced by the density of states (DOS) near the Fermi level in Figure S11b. At the domain interface, additional oxygen leads to a high-energy shift of the O $2p$ orbital, amplifying Ni-O hybridization (Figure S11c). Similar pyramidal distortions at the $LaGaO_3$/$LaNiO_2$ interface have been shown to confine hole doping to the interface region[7]. In addition, linear dichroic resonant x-ray reflectometry demonstrated a fractional formation of $a$-axis-oriented infinite layer structure in the dominant $c$-axis infinite layer $NdNiO_2$ phase in the $8NdNiO_2/4SrTiO_3$ superlattice[26]. Another configuration (Figure S12) combines the $c$-axis-oriented infinite-layer structure and the perovskite phase, showing a disordered distribution of residual oxygen. The DOS of the $c$-axis infinite layer structure in Figure S12c exhibits a slight enhancement in Ni–O hybridization near the Fermi level compared to that shown in Figure S11d. In contrast, the perovskite phase displays pronounced distortions of the $NiO_6$ octahedra, with multiple local configurations identified (Figure S12a). This difference arises from the polarity between the infinite layer and the perovskite structures. Correspondingly, Figures S12b and S12d reveal a difference in Ni–O hybridization, underscoring the intricate interplay between residual oxygen, structural distortions, and the resulting electronic complexity.

## Discussion

In practical experiments, weak defocus can significantly affect the quantification of atomic occupancy derived from iCoM phase contrast imaging[29]. This limitation is especially pronounced in complex oxides with low atomic occupancy, where depth information about the atomic structure may be overlooked. In contrast, phase-contrast

images reconstructed using multi-slice ptychography at different depths directly resolve the residual oxygen at apical sites within the infinite-layer nickelate structure. The role of residual oxygen in enabling hole doping has been proposed as a possible explanation for the emergence of superconductivity in un-doped infinite-layer nickelates single film[10]. However, the inconsistent emergence of superconductivity in single infinite-layer nickelate films—present in some cases[10,11] and absent in others[1], despite seemingly similar growth conditions—suggests that residual oxygen may play a more complex and nuanced role in these systems. Our detailed analysis of the oxygen sublattice reveals that residual apical oxygen is distributed in a disordered manner along the depth direction. Furthermore, basal oxygen exhibits depth-dependent distortions, reminiscent of the structural instabilities previously reported for infinite-layer nickelates[30,31].

Comparatively, the absence of a pre-peak around ~527.5 eV in the O $K$-edge for infinite-layer nickelates, as opposed to its presence in the perovskite phase of $NdNiO_3$, highlights the weakened hybridization between Ni $3d$ and O $2p$ orbitals in the $Ni^+$ state[21]. This characteristic is predominant across our sample, confirming the presence of the infinite-layer structure. However, a weak pre-peak at slightly higher energy suggests that hole doping in the infinite-layer structure related to residual oxygen, akin to observations of hole doping via A-site cation substitutions in other nickelates[21]. The sporadic appearance of this feature implies an inhomogeneous hole-doping effect, likely associated with the residual apical oxygen observed in our depth-resolved phase contrast images.

The disordered residual oxygen likely contributes to the formation of buried pyramidal and octahedral structures within the infinite-layer nickelate. The DFT calculations demonstrate that the region with residual oxygen can form distinct pyramidal or octahedral distortion in the case of a sharp decrease in the out-of-plane lattice after the deintercalation of the apical oxygen. This structural modification significantly impacts the electronic structure near the Fermi level by enhancing Ni-O hybridization. In particular, pyramidal distortions can modify the Nd-Ni hybridization near the Fermi level. Additionally, pyramidal structures form at the interface between $a$-axis- and $c$-axis-oriented infinite-layer domains. The $a$-axis-oriented domains exhibit stronger Nd-Ni hybridization compared to their $c$-axis counterparts, potentially amplifying of self-hole doping effects that are conducive to superconductivity[32]. The proportion of $c$-axis- and $a$-axis-oriented domains emerges as a critical parameter for tuning the electronic states across the entire film.

In summary, depth-resolved phase contrast imaging reconstructed via multi-slice ptychography highlights the presence of disordered residual oxygen in infinite-layer nickelates. Such oxygen is often underestimated in iCoM images, particularly under weak defocus conditions. Our combined STEM-EELS analysis reveals the hole doping locally in the infinite inner layer. Additionally, pyramidal distortions at the junctions of $a$- and $c$-axis-oriented domains play a similar role in contributing holes to the system. Although superconductivity was not observed in our infinite-layer nickelate superlattice, this study underscores the importance of residual oxygen and its structural effects. Achieving precise control over the level and distribution of residual oxygen remains a significant challenge. Future investigations combining advanced depth detection methods, such as multi-slice ptychography, with high-resolution EELS or complementary techniques, will be essential to disentangle the individual contributions of residual oxygen, domain interfaces, and their relative proportions to the electronic properties of these materials.

## Methods

### Materials and Sample Preparation

An $8NdNiO_3/2SrTiO_3$ superlattice was epitaxially grown on a (001)-oriented single-crystal $SrTiO_3$ substrate by pulsed laser deposition (PLD) technique, following the same procedures described in references[7,8], using a KrF excimer laser with 2-Hz pulse rate and 1.6-J/$cm^2$ energy density. The superlattice film was deposited in 0.5-mbar oxygen atmosphere at 730 °C and then annealed in 1-bar oxygen atmosphere at 690 °C for 30 min. The resulting structure underwent a topotactic reduction process using calcium hydride ($CaH_2$) powder in a vacuum-sealed Pyrex glass tube at 280 °C for approximately 4.5 days.

Transmission electron microscopy (TEM) lamellae were prepared using a focused ion beam (FIB Scios, FEI) system under a high-vacuum environment. To mitigate beam-induced charging effects during FIB milling, a protective carbon layer of approximately 6 nm thickness was deposited onto the sample surface using a high-vacuum sputter coater (EM ACE 600, Leica). The final quality of the TEM lamellae was further enhanced using a Fischione NanoMill® TEM sample preparation system, which employed low-energy ion milling and high-vacuum cleaning to reduce damage and contamination.

### STEM Imaging and STEM-EELS acquirements

Scanning transmission electron microscopy (STEM) investigations were carried out using a JEOL JEM-ARM200F microscope (JEOL Co. Ltd.) equipped with a DCOR probe corrector and a Gatan GIF Quantum ERS K2 spectrometer. High-angle annular dark field (HAADF) imaging utilized a 30 μm condenser aperture, corresponding to a convergence semi-angle of 20.4 mrad. The collection semi-angle ranged from 83 to 205 mrad for HAADF imaging and was set to 85 mrad for electron energy loss spectroscopy (EELS) measurements. EELS spectra were acquired with a dispersion of 0.5 eV/channel, achieving an energy resolution of approximately 1 eV. For 4D-STEM data acquisition, a Merlin pixelated detector (256 × 256 pixels, Quantum Detectors) operating in 1-bit mode was employed, enabling continuous read/write at a pixel dwell time of 48 μs. The probe current was approximately 23 pA. The convergent angle is semi-angle of 20.4 mrad. The scan step size is approximately 0.17 Å, and the corresponding electron dose is approximately $2.4 \times 10^5 \, e^-/Å^2$. The angular sampling in the diffraction patterns is approximately 0.376 mrad/pixel. The defocus was approximately 50 Å, resulting in a probe overlap ratio of approximately 72% for the reconstruction. The sliced phase contrast images at different depths (Figures S2 and S3) and the intensity of all the probe modes (Figure S13) were used to verify the reliability of the reconstruction. Post-processing of the 4D-STEM datasets was conducted using Python-based libraries, specifically py4DSTEM[33] and fpd[34], to reconstruct ADF and iCoM images. For the 4D-STEM image simulation, we used multislice electron scattering algorithm, which was based on the abTEM python library[35]. We performed the mixed-state multislice ptychography reconstruction on the experimental and simulated results using the open-source python package of py4DSTEM[33] and the open-source MATLAB package of PtychoShelves[19,36–38].

### DFT Calculations

Density functional theory (DFT) calculations were performed to explore the structural distortions and density of states (DOS) variations in the infinite-layer nickelate within the $8NdNiO_2/2SrTiO_3$ superlattice, accounting for residual oxygen. These calculations employed the Generalized Gradient Approximation (GGA) with the Perdew-Burke-Ernzerhof (PBE) functional for exchange-correlation, as implemented in the Vienna Ab initio Simulation Package (VASP)[39,40]. A plane-wave cut-off energy of 520 eV was used, and the DFT + U method included a Hubbard U parameter of 4.0 eV for Ni atoms to accurately describe the localized electronic states. Structural optimization was conducted with stringent criteria, setting the maximum ionic force to 0.01 eV/Å, self-consistent total energy convergence to $5 \times 10^{-7}$ eV/atom, and the maximum ionic displacement to 0.5 Å. Analysis of the VASP-generated data was facilitated using the VASPKIT toolkit[41].

## Data availability

The data supporting the findings of this study are available in the main text and Supplementary Information. The source data file and 4D-STEM raw data have been deposited on the Figshare database under the accession code https://doi.org/10.6084/m9.figshare.30581966.

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

## Acknowledgements

This project has received funding from the European Union's Horizon 2020 research and innovation programme under Grant Agreement No. 823717-ESTEEM3. We are thankful to Dr. T. Heil for the support of merlin software, K.H. and P.K. for TEM support.

## Author contributions

C.Y. conceived the project. C.Y., H.G.W., and P.A.v.A. conducted the STEM measurements and related data analysis. R.A.O. grew the samples and performed the topotactic reduction. P.A.v.A, E.B, and B.K. supervised this work. C.Y. did the simulations and calculations. K.A. provided

useful help of the DFT calculations. C.Y. wrote the paper with contributions from all authors. All authors contributed to the discussion and comments.

## Funding

## Competing interests
The authors declare no competing interests.
