## [Transparent Peer Review file · Nature Communications]

Direct imaging of residual oxygen disorder in an infinite-layer nickelate superlattice via multislice ptychography

Corresponding Author: Dr Chao Yang

Version 0:

Reviewer comments:

Reviewer #1

(Remarks to the Author)

The manuscript by Yang, et al. thoroughly demonstrates the use of multislice ptychography to image residual oxygen in infinite layer nickelate superlattices and discusses the significant implications for our understanding of superconductivity in these materials. They support their discussion with EELS analysis of the O-K edge spectra, showing inhomogeneous residual oxygen and DFT calculations, showing the potential effects on electronic structure. It is interesting and well-supported work with broad applications in materials science beyond superconducting nickelates. However, I have a few concerns that should be addressed to improve the manuscript prior to publication:

1. The authors do not mention the limits of depth (z) resolution in multislice ptychography. Although I don't believe this poses a problem for the conclusions drawn in this study, the depth resolution limitations should be mentioned and connected to the present oxygen analysis.
2. The authors should specify the multislice ptychography and multislice image simulation algorithms they used.
3. In Figures 2, 3, and S4 it seems to me, visually, that there are more cases of apical oxygen contrast increasing with sample depth as opposed to the opposite. Is this the case across other datasets? If so, I'm concerned this could be an artifact of the reconstruction.
4. Line 167-168: While the NdNiO₂/SrTiO₃ interfaces do appear quite sharp, the elemental maps (Fig. 4b) do show one atomic layer of intermixing between the Ni and Ti in the lower interface.
5. The authors should discuss the motivation for including a-axis-oriented IL structures in their DFT calculations at the beginning of that section. Up to that point in the manuscript, there is no mention of different IL orientations, making the inclusion of the a-axis orientation in the DFT discussion feel somewhat abrupt and incongruous. Figure S8 is a more logical addition to the manuscript, as it addresses residual perovskite phase character. However, if Figure S8a represents the full supercell used in the calculations, I don't think it's a very good representation of the disordered residual oxygen observed experimentally. This should be addressed at least with additional text and perhaps with additional calculations.
6. Line 217-218: This statement should be supported with a reference or additional data.

Reviewer #2

(Remarks to the Author)

This study investigates residual oxygen disorder in infinite-layer nickelate superlattices, employing multislice ptychography to directly image low-occupancy oxygen sites. The findings reveal that residual oxygen induces structural distortions and inhomogeneous hole doping, as confirmed by EELS and supported by DFT calculations. The critical data lies in visualizing oxygen disorder by multislice ptychography, and further discuss the interplay between oxygen vacancies, structural distortions, and electronic properties. However, superconductivity was not observed in the studied superlattice, making the

relative discussion and mechanism unconvincing. In addition, the existence of residual apical oxygen in infinite-layer nickelate films is widely recognized. The primary innovation of this research lies in the successful implementation of multislice ptychography for high-resolution imaging of apical oxygen atoms in infinite-layer nickelates. In the following, I have several questions on the STEM data and main conclusions, before the consideration of its recommendation.

1. Although multislice ptychography has advantages in visualizing oxygen atoms with low occupancy, the omission of quantitative analysis undermines the study's scientific robustness and broader implications. The study primarily relies on qualitative observations of oxygen distribution without quantitative analysis. Including statistical data on oxygen occupancy and its correlation with electronic properties would strengthen the findings.

2. The experimental section lacks detailed descriptions of the 4D-STEM experiments, including the parameters of collecting data such as convergence angle, scanstep, defocus, the sampling in the diffraction patterns and electron dose, etc., and parameters during the reconstructed process. This makes it difficult to evaluate the reliability of the phase contrast images. For example, intensity of all the probe modes from mixed-state algorithm should be provided in the supplementary to verify the reliability of the ptychography reconstruction.

3. I am curious on Figure 2 that the quality of the multi-slice ptychography phase-contrast image is obviously worse than that of the iCOM image, because the multi-slice ptychography should perform with a higher resolution and image contrast. The origin of the elongation contrast of each atomic column in Fig.2e and Fig.3b should be explained, to avoid misleading in the identification on the residual oxygen.

4. In the simulated phase contrast image in Fig. S2, why the phase contrast on atomic columns is big and diffused at the depth range from 4 to 60 angstrom but converged at the depth of 100 angstrom. Similar situation appears in Fig.S4. Some further evidences are needed to verify the successful reconstruction.

5. The EELS spectra are presented, but their interpretation in relation to hole doping and superconductivity is vague. The manuscript draws conclusions about the role of residual oxygen in enabling superconductivity, but these are not fully supported by the presented data. The link between oxygen disorder and superconductivity remains speculative.

Reviewer #3

(Remarks to the Author)
see attachment

Reviewer #4

(Remarks to the Author)

Version 1:

Reviewer comments:

Reviewer #1

(Remarks to the Author)

I appreciate the authors' efforts in revising the manuscript. They have satisfactorily addressed all the issues raised in the first round of review.

Reviewer #2

(Remarks to the Author)

The authors have addressed all my concerns, and the manuscript is now substantially improved. I think it is acceptable.

Reviewer(s)' Comments to the Author

Reviewer #1 (Remarks to the Author):

The manuscript by Yang, et al. thoroughly demonstrates the use of multislice ptychography to image residual oxygen in infinite layer nickelate superlattices and discusses the significant implications for our understanding of superconductivity in these materials. They support their discussion with EELS analysis of the O-K edge spectra, showing inhomogeneous residual oxygen and DFT calculations, showing the potential effects on electronic structure. It is interesting and well-supported work with broad applications in materials science beyond superconducting nickelates. However, I have a few concerns that should be addressed to improve the manuscript prior to publication:

1. The authors do not mention the limits of depth (z) resolution in multislice ptychography. Although I don't believe this poses a problem for the conclusions drawn in this study, the depth resolution limitations should be mentioned and connected to the present oxygen analysis.

Response: Thank the reviewer for your insightful comment. We have revised the manuscript to include the discussion and simulated results related to depth resolution. Specifically, the depth resolution using multislice ptychography has been demonstrated to be about 2.0 nm for oxygen dopants within a 15-nm-thick PrScO_3 structure. (Ref., Science 2021, 372, 826-831.) The simulation of the 10-nm-thick NdNiO_{2+x} structure demonstrates a comparable depth resolution of approximately 2.3 nm for oxygen atoms, as shown in the Figure R1. Figure R1a shows the projected phase contrast image with 15% apical oxygen occupancy. The depth profile in Figure R1b has been derived from the yellow line across oxygen and Nd atoms in Figure R1a, showing the distribution of residual oxygen in the depth direction. This is agreement with the structural model in Figure R1c. The multislice phase contrast images in different depths are shown in Figure R1d, where the residual apical oxygen is visible. The depth resolution of the experimental result can be slightly worse due to the effects of noise signal, beam instabilities, and the sample quality. As also the reviewer mentioned, the depth resolution here is sufficient to measure the residual oxygen in our sample.

Figure R1. (a) A projected phase contrast image of a 10-nm-thick NdNiO_{2+x} structural model. (b) The depth profile plot for oxygen and Nd atoms extracted from the yellow dashed line in (a). The elliptical dashed shapes mark the residual apical oxygen in the depth direction. (c) The corresponding structural model with the distribution of the residual apical oxygen. (d) Reconstructed phase contrast images at different depths.

2. The authors should specify the multislice Ptychography and multislice image simulation algorithms they used.

Response: We thank the reviewer for this valuable comment. In the revised manuscript, we have included the details of the multislice image simulation algorithms in the method section. Specifically, we employed the gradient-descent method for the mixed-state multislice Ptychography reconstruction of the experimental results as described in the open-source Python package of py4DSTEM (Ref., *Microscopy and Microanalysis* 2021, 27, 712-743) and we performed the mixed-state multislice Ptychography reconstruction of the simulated results using the open-source MATLAB package of PtychoShelves (Refs., *Science* 2021, 372, 826-831; *Nature* 2013, 494, 68-71; *J. Appl. Cryst.* 2020, 53, 574; *Opt. Express* 2016, 24, 29089-29108.). For the simulation of the 4D dataset, we used a multislice

electron scattering algorithm based on the open-source package of abTEM (Ref., Open Research Europe 2021, 1, 24).

3. In Figures 2, 3, and S4 it seems to me, visually, that there are more cases of apical oxygen contrast increasing with sample depth as opposed to the opposite. Is this the case across other datasets? If so, I'm concerned this could be an artifact of the reconstruction.

Response: Thank the reviewer for the insightful comment. As shown in Figure R1, we have created a model with residual apical oxygen located in the middle of a 10-nm-thick NdNiO_{2+x} structural model. The residual oxygen can be imaged in the sliced phase contrast images. We have checked more datasets from different regions, showing different configurations of residual oxygen. To avoid possible artefacts in the top and bottom layers of the real sample, we extracted phase-contrast images from middle slices with good reconstruction to show the residual oxygen (See Figures R2-R4). Figure R2 shows the interface region, from which extracted slices of the phase-contrast images were obtained at depths of 45 Å and 75 Å, respectively. The projected phase contrast image in Figure R2c was obtained by summing the middle slices after the top and bottom four slices were removed. The same processing was performed on Figures R4 and R5. Distinct residual apical oxygen, marked with white dashed circles in Figure R2e, is observed in the phase-contrast image at a depth of 75 Å, but not at a depth of 45 Å, as shown in Figure R2d. Figure R3 shows the complete reconstructed slices. The reconstruction of the upper layers was found to be suboptimal, a phenomenon that is likely associated with the decomposition of the surface phase that occurred during the chemical reduction procedure. Furthermore, the surface layer of the reduced sample exhibits strong reconstruction, as previously reported in our research (Ref., *Nat. Commun.*, 2024, 15, 378). This has the potential to increase the complexity of the phase shift in the surface region. However, the presence of residual oxygen in the middle slices of the sample has been confirmed.

In the case of the NdNiO₂ inner layer region shown in Figure R4c, the presence of apical oxygen is not discernible in the projected phase contrast image, indicating an infinite layer structure. However, Figure R3e demonstrates the presence of residual apical oxygen at 80 Å depth. This is not the case at a depth of 45 Å in Figure R3d. Furthermore, Figure R5 shows an additional NdNiO₂ inner layer region. The presence of residual oxygen is evident in the projected phase contrast image (Figure R5b), marked with a white dashed circle, which forms a pyramid NiO₅ structure. The residual apical oxygen is visible at depths of 35 Å and 50 Å, but its contrast noticeably decreases at 80 Å. Therefore, these different configurations of residual oxygen in the infinite layer nickelate sample demonstrate the disordered nature of its distribution. In response to the reviewer's concerns, we have added the related results and discussion to the supplementary information and the manuscript.

Figure R2. (a) Reconstructed annular dark field (ADF) image obtained from the 4D-STEM dataset, providing the atomic structure of the $8\text{NdNiO}_2/2\text{SrTiO}_3$ superlattice sample. (b) A magnified ADF image of the $\text{SrTiO}_3/\text{NdNiO}_2$ interface region, outlined by the yellow dashed box in (a). (c) The corresponding projected phase-contrast image reconstructed using multi-slice ptychography. Phase-contrast images at depths of (d) 45 \AA and (e) 75 \AA . The white dashed circles indicate the location of the residual apical oxygen.

Figure R3. Experimental phase contrast images of NdNiO_{2+x} showing all slices in different depths. The white circle shows the evolution of the apical oxygen contrast in different depths.

Figure R4. (a) Reconstructed annular dark field (ADF) image obtained from the 4D-STEM dataset, providing the atomic structure of the $8\text{NdNiO}_2/2\text{SrTiO}_3$ superlattice sample. (b) A magnified ADF image of the NdNiO_2 inner layer region, outlined by the yellow dashed box in (a). (c) The corresponding projected phase-contrast image reconstructed using multi-slice ptychography. Phase-contrast images at depths of (d) 50 \AA and (e) 80 \AA . The white dashed circles indicate the location of the residual apical oxygen.

Figure R5. (a) Reconstructed annular dark field (ADF) image obtained from the 4D-STEM dataset, providing the atomic structure of the $8\text{NdNiO}_2/2\text{SrTiO}_3$ superlattice sample. (b) The projected phase-contrast image of the NdNiO_2 inner layer region were reconstructed using multi-slice ptychography, and are outlined by the yellow dashed box in (a). Phase-contrast images at depths of (c) 35 \AA , (d) 50 \AA , and (e) 85 \AA . The white dashed circles indicate the location of the residual apical oxygen.

4. Line 167-168: While the $\text{NdNiO}_2/\text{SrTiO}_3$ interfaces do appear quite sharp, the elemental maps (Fig. 4b) do show one atomic layer of intermixing between the Ni and Ti in the lower interface.

Response: Thank the reviewer for the valuable comment. We are sorry for this misleading description. Indeed, the interface is sharp from the contrast of HAADF image while there is still one unit cell intermixing from EELS measurement. We have corrected this description in the manuscript.

5. The authors should discuss the motivation for including a-axis-oriented IL structures in their DFT calculations at the beginning of that section. Up to that point in the manuscript, there is no mention of different IL orientations, making the inclusion of the a-axis orientation in the DFT discussion feel somewhat abrupt and incongruous. Figure S8 is a more logical addition to the manuscript, as it addresses residual perovskite phase character. However, if Figure S8a represents the full supercell used in the

calculations, I don't think it's a very good representation of the disordered residual oxygen observed experimentally. This should be addressed at least with additional text and perhaps with additional calculations.

Response: Thank the reviewer for the constructive suggestion. We have included the motivation of DFT calculations at the beginning of that section in the manuscript at page 6, as below “Given the uneven removal of oxygen ligands during the reduction process by annealing with CaH_2 , a twin structure consisting of *a*-axis oriented and *c*-axis oriented infinite layers can form. (Cryst. Growth Des. 2010, 10, 5, 2044–2046.) The *a*-axis oriented infinite layer structure was dominated in the $\text{NdNiO}_2/\text{SrTiO}_3$ superlattice, as shown by *X*-ray resonant reflectometry (26) To further elucidate the possible effects of disordered residual oxygen in our sample, we performed DFT calculations to model various configurations”

We thank the reviewer for their recognition and insightful comment regarding the configuration of the residual perovskite phase shown in Figure S8. To more accurately model the disorder residual oxygen in the infinite layer phase, we have added the supercell structure with more perovskite unit cells embedded within infinite layer phase in Figure R6. It demonstrates a disordered distribution of residual oxygen. In the *c*-axis oriented infinite layer structure, the DOS (See Figures R6c) exhibits a slight enhancement in Ni–O hybridization near the Fermi level compared to that in Figure S11d. In contrast, the perovskite phase displays pronounced distortions of the NiO_6 octahedra, with multiple local configurations identified (Figure R6a). This difference arises from the polarity between the infinite layer and the perovskite structures. Correspondingly, Figures R6b and R6d reveal a difference in Ni–O hybridization, underscoring the intricate interplay between residual oxygen, structural distortions, and the resulting electronic complexity. The corrections have been incorporated into the manuscript.

Figure R6. (a) Supercell model comprising the infinite-layer structure (c) together with disordered residual perovskite units (b, d). The corresponding density of states is presented for panels (b), (c), and (d).

6. Line 217-218: This statement should be supported with a reference or additional data.

Response: Thank the reviewer for the valuable comment. We have included a reference to the effect of focus on the phase contrast of the iCoM image. C. Gao et al (Ref., *Ultramicroscopy* 2024, 256, 113879) conducted a detailed study of effects of defocus on the contrast of the iCoM image. The iCoM image demonstrates optimal contrast when the electron probe is focused on the center of the sample. It is evident that significant variations in probe defocus have a substantial impact on phase contrast.

Reviewer #2 (Remarks to the Author):

This study investigates residual oxygen disorder in infinite-layer nickelate superlattices, employing multislice ptychography to directly image low-occupancy oxygen sites. The findings reveal that residual oxygen induces structural distortions and inhomogeneous hole doping, as confirmed by EELS and supported by DFT calculations. The critical data lies in visualizing oxygen disorder by multislice ptychography, and further discuss the interplay between oxygen vacancies, structural distortions, and electronic properties. However, superconductivity was not observed in the studied superlattice, making the relative discussion and mechanism unconvincing. In addition, the existence of residual apical oxygen in infinite-layer nickelate films is widely recognized. The primary innovation of this research lies in the successful implementation of multislice ptychography for high-resolution imaging of apical oxygen atoms in infinite-layer nickelates. In the following, I have several questions on the STEM data and main conclusions, before the consideration of its recommendation.

1. Although multislice ptychography has advantages in visualizing oxygen atoms with low occupancy, the omission of quantitative analysis undermines the study's scientific robustness and broader implications. The study primarily relies on qualitative observations of oxygen distribution without quantitative analysis. Including statistical data on oxygen occupancy and its correlation with electronic properties would strengthen the findings.

Response: We thank the reviewer for the insightful comment. In response, an attempt has been made to provide a more quantitative analysis of the results. The simulation results in Figure 1 indicate that the estimation of the oxygen occupancy of the residual oxygen from the phase contrast image is consistent with the structural model. Furthermore, an ideal multislice ptychography reconstruction has been demonstrated that the phase of the complex object function is linearly proportional to the electrostatic potential from the atoms. (Ref., *Nature* 2024, 630, 847–852.)

However, the quantification of experimental results remains challenging due to the complexity of the experimental sample. In particular, the reduction process of the nickelate sample introduces additional phase shifts in the surface region, likely caused by surface phase decomposition (Ref., *Nano Lett.* 2023, 23, 3291–3297) and strong surface reconstruction due to polarity effects (Ref., *Nat. Commun.* 2024, 15, 378). However, as requested by the reviewer, a statistical estimation of the oxygen occupancy was performed based on the projected phase contrast image, as shown in Figure R7. The oxygen sites in SrTiO₃ are regarded as being fully occupied and can serve as the reference for 100% occupancy. The definition of 0% oxygen occupancy is challenging due to the presence of residual contrast at the oxygen sites. Assuming that the minimum observed phase contrast corresponds to 0% oxygen occupancy, Figure R7d suggests that residual oxygen occupancy is mostly in the range of 10–15%.

Figure R7. (a) Projected phase-contrast image reconstructed using multi-slice ptychography. The normalized phase contrast map of (b) the basal oxygen and (c) the apical oxygen extracted from (a). (d) Histogram of phases from basal and apical oxygen corresponding to (b) and (c), respectively.

Figure R8. Simulated phase contrast images by multislice ptychography reconstruction with different oxygen occupancy (4%, 12%, 23%, 35%, and 50%) at the apical sites.

To avoid overstating the precision of these estimations, it is preferable to offer a range of possible oxygen occupancies in the results. In most infinite-layer regions, the residual apical oxygen is invisible in both the iCoM image and the projected phase contrast image, becoming detectable only in sliced phase-contrast images. This has been confirmed across multiple residual oxygen configurations at both the interface and within the infinite-layer phase (Figures R2–R5). In this case of the invisible apical oxygen contrast in infinite layers, the oxygen occupancy in the infinite layer nickelates is below 12% according to the simulation results in Figure R8. This is also within the range estimated in Figure R7. The estimations of the oxygen occupancy correspond to the Ni^{-8.8} electronic configuration of NdNiO_{2.12} or an even lower oxidation state of Ni. The Ni^{-8.8} electronic configuration with hole-doping through Sr dopants has been shown to exhibit the phenomenon of superconductivity, which was absent in the

undoped infinite layer nickelate single films (Ref., Nature 2019, 572, 624–627). Recent studies have demonstrated the realization of superconductivity in infinite-layer nickelate single films (Refs., arXiv:2410.02007, 2024; arXiv:2410.16147, 2024.). However, the role of residual oxygen remains unclear. Our findings suggest that disordered residual oxygen modifies the electronic structures in multiple configurations and potentially affecting the self-hole doping effects that are conducive to the phenomenon of superconductivity. The results of the oxygen occupancy estimation have been included in the supplementary information and the related discussion in the manuscript.

2. The experimental section lacks detailed descriptions of the 4D-STEM experiments, including the parameters of collecting data such as convergence angle, scanstep, defocus, the sampling in the diffraction patterns and electron dose, etc., and parameters during the reconstructed process. This makes it difficult to evaluate the reliability of the phase contrast images. For example, intensity of all the probe modes from mixed-state algorithm should be provided in the supplementary to verify the reliability of the ptychography reconstruction.

Response: We thank the reviewer for the insightful and rigorous comment. The description of the experimental parameters has been incorporated into the method section. In particular, the convergence angle of 20.4 mrad, defocus of approximately 50 Å, scanstep size of approximately 0.17 Å, electron dose of approximately $2.4 \times 10^5 \text{ e}^-/\text{Å}^2$, and the angular sampling in the diffraction patterns of approximately 0.376 mrad/pixel. A probe overlap ratio of approximately 72% provides sufficient redundancy for the reconstruction. In addition, in order to evaluate the reliability of the phase contrast images, the simulation results in Figures R1 and R11 have been added and recalculated. The sliced phase-contrast images presented in these figures evidently demonstrate the distribution of residual oxygen at varying depths. As recommended by the reviewers, the intensity of all the probe modes as shown in Figure R9 from the mixed-state algorithm has been incorporated into the supplementary information. The probe intensity is predominantly concentrated in the first few modes, indicating a well-constrained and effective reconstruction. The quality of the reconstructions can also be checked by examining the sliced phase contrast images from the simulation and experiments.

Figure R9. Intensity of all the probe modes from mixed-state algorithm. Indexes of modes and the corresponding fractional intensity in the total incident beam are labeled on the probe intensity images.

3. I am curious on Figure 2 that the quality of the multi-slice ptychography phase-contrast image is obviously worse than that of the iCOM image, because the multi-slice ptychography should perform with a higher resolution and image contrast. The origin of the elongation contrast of each atomic column in Fig.2e and Fig.3b should be explained, to avoid misleading in the identification on the residual oxygen.

Response: Thank the reviewer for the insightful comment. The multi-slice ptychography phase-contrast images in Figure 2 have been modified as outlined below (Figure R10) to demonstrate enhanced contrast of the oxygen sublattice. The observed elongation of the atomic column, particularly in the inner layer of infinite-layer nickelate, is attributed to residual oxygen-induced sublattice distortion. At the interface, the strong interface polarity induces a shift of the apical oxygen towards the Nd columns. In contrast, this elongation contrast is not evident in the SrTiO₃ layer. The related discussion has been added to the paragraph of Figure 2.

Figure R10. Comparison of experimental visualization of the oxygen sub-lattice through reconstructed iCoM and multi-slice ptychography phase-contrast images, highlighting the direct imaging of residual apical oxygen along the z -direction in the $8\text{NdNiO}_2/2\text{SrTiO}_3$ super-lattice film. (a) Reconstructed annular dark field (ADF) image. (b) iCoM image derived from the 4D-STEM dataset. (c) Magnified ADF image of the $\text{SrTiO}_3/\text{NdNiO}_2$ interface region, outlined by the red dashed box in (a). (d) Corresponding magnified iCoM image of the same region. (e) Phase-contrast image of the $\text{SrTiO}_3/\text{NdNiO}_2$ interface reconstructed using multi-slice ptychography, with accompanying depth profiles of (f) basal oxygen and (g) apical oxygen. (h) Magnified ADF image of the NdNiO_2 inner layer region, outlined by the yellow dashed box in (a). (i) Corresponding magnified iCoM image of the NdNiO_2 inner layer. (j) Phase-contrast image of the NdNiO_2 inner layer region reconstructed using multi-slice ptychography, along with the depth profiles of (k) basal oxygen and (l) apical oxygen.

4. In the simulated phase contrast image in Fig. S2, why the phase contrast on atomic columns is big and diffused at the depth range from 4 to 60 angstrom but converged at the depth of 100 angstrom. Similar situation appears in Fig.S4. Some further evidences are needed to verify the successful reconstruction.

Response: We thank the reviewer for the insightful comment. The phase contrast diffusion phenomenon is likely attributable to the reconstruction process not having fully converged for the heavy atoms located in the uppermost layers. It has been demonstrated that the heavier atoms exhibit a greater phase shift in comparison to the lighter atoms, thereby resulting in a slight diffusion contrast. The reconstruction of phase contrast images has been modified through the execution of additional iterations to achieve enhanced convergence of the phase contrast, as shown in Figure R11. The corrected phase contrast images demonstrate a sufficient convergence of the phase contrast. Furthermore, Figure R1 demonstrates a different configuration in which residual oxygen is located centrally within the sample.

The reconstructed phase contrast image aligns closely with the structural model setting. The corresponding Figures S2 and S4 in the manuscript have been modified, as shown in Figures R11 and R3, respectively.

Figure R11. Simulated phase contrast images of 10-nm-thick NdNiO_{2+x} with 23% occupancy of apical oxygen showing all slices in different depths.

5. The EELS spectra are presented, but their interpretation in relation to hole doping and superconductivity is vague. The manuscript draws conclusions about the role of residual oxygen in enabling superconductivity, but these are not fully supported by the presented data. The link between oxygen disorder and superconductivity remains speculative.

Response: We thank the reviewer for the valuable comment. We agree that establishing a clear connection between hole doping and superconductivity is essential. However, despite significant efforts in the field, the achievement of superconductivity in infinite-layer nickelates has not yet been achieved through hole doping via the superlattice approach. This limitation hinders a direct link of the role of hole doping in enabling superconductivity in superlattice samples. Furthermore, the observation that

superconductivity appears in some single infinite-layer nickelate films (Refs., arXiv:2410.02007, 2024; arXiv:2410.16147, 2024.) but not in others (Ref., Nature 2019, 572, 624–627)—despite nominally similar growth conditions across different research groups—demonstrates the complex role of residual oxygen in these systems. In response to the reviewer’s comment, we have revised the manuscript to more explicitly describe the presence of disordered residual oxygen, its potential to induce hole doping, and the possible consequences for the electronic structure in infinite-layer nickelates. Furthermore, we have carefully checked and rephrased our conclusions to avoid overstating any direct causal relationship between oxygen disorder, hole doping, and superconductivity in our samples. Instead, we now emphasize that our EELS data indicate modifications in oxygen coordination and electronic structure, which may induce hole doping. This is a factor that is known to be important for superconductivity. We believe these revisions render our conclusions more balanced and better supported by the experimental evidence.

Reviewer #3 (Remarks to the Author):

This work reported distorted basal O and apical O distribution along the projection direction using multi-slice ptychography. EELS is complemented to evidence the hole doping of those apical O. I have some questions on the current conclusions and suggestions on developing this paper.

1. Figure 1 is purely on simulated data, this means there is a theoretical model first, then images are simulated based on this. Reading the paragraph of figure 1 indicates a reverse way of expression. I believe the purpose of figure 1 is to demonstrate the apical O with 35% occupancy can be imaged in this model. This model is not clearly described. It seems apical O concentrates at the bottom region. Figure 1a does not show an over-defocus probe.

Response: We thank the reviewer for the insightful comment and suggestion. We have added another supplemental simulation with a different structural model of 10-nm-thick NdNiO_{2+x} , in which the residual oxygen is located solely in the middle, as shown in Figure R1. Figure R1a shows the projected phase contrast image with 15% apical oxygen occupancy. The depth profile in Figure R1b has been derived from the yellow line across oxygen and Nd atoms in Figure R1a, showing the distribution of residual oxygen in the depth direction. This is agreement with the structural model in Figure R1c. The multislice phase contrast images in different depths are shown in Figure R1d, where the residual apical oxygen is visible. This provides further information about the different configurations of residual oxygen in this system. We have added the corresponding text to the paragraph of Figure 1. This helps to highlight the disordered residual oxygen in our samples. Additionally, we have modified the defocus probe in Figure 1a.

Figure R1. (a) A projected phase contrast image of a 10-nm-thick NdNiO_{2+x} structural model. (b) The depth profile plot for oxygen and Nd atoms extracted from the yellow dashed line in (a). The elliptical dashed shapes mark the residual apical oxygen in the depth direction. (c) The corresponding structural model with the distribution of the residual apical oxygen. (d) Phase contrast images were reconstructed at varying depths.

2. Figure S2 shows very abnormal reconstruction at each slice. The question is why the top slices look like not convergent, although there is a defocus.

Response: We thank the reviewer for the insightful comment and suggestion. During phase reconstruction, the exit electron wave carries more information about the deeper slices than the top layers during the phase reconstruction, since the electrons scatter forward from the top to the bottom of the sample. Convergence is observed to be slower for the upper slices in comparison to the lower slices. As the reviewer suggested, we have checked and modified the reconstruction of the phase contrast image by performing more iterations to achieve better convergence of phase contrast for the top layers, as shown in Figure R11. The corresponding Figure S2 in the manuscript has been modified.

Figure R11. Simulated phase contrast images of 10-nm-thick NdNiO_{2+x} with 23% occupancy of apical oxygen showing all slices in different depths.

3. The authors claim: The discrepancy observed between the phase contrast and the structural model is closely related to sample thickness, and the scattering angles and defocus conditions of the reconstruction, amongst other factors. I do not think they are suitable here, the key reason should be due to the contrast transfer function of the phase methods and the capabilities of coping with the dynamic scattering effects.

Response: We appreciate the reviewer for the insightful comment and suggestion. We agree that the contrast transfer function of the phase reconstruction methods and their ability to handle dynamical scattering effects are indeed key factors contributing to the observed discrepancy between the phase contrast and the structural model. Sample thickness, scattering angles, and defocus conditions are some of the related experimental factors that affect the reconstruction. We have corrected the statements in the manuscript.

4. From figure 2, how the authors conclude the apical O has 12% ?

Response: Thank the reviewer for the insightful question. As shown in the simulation results in Figure R8 and S1, residual apical oxygen becomes indistinguishable in the projected phase contrast images and iCoM image when its occupancy falls below 12%. However, this can still be resolved in the depth-sliced phase contrast images. In view of the present sample quality and the existence of spatially disordered residual oxygen, it is difficult to determine an exact occupancy value. Furthermore, a statistical estimation of the oxygen occupancy was performed based on the projected phase contrast image, as shown in Figure R7. The oxygen sites in SrTiO₃ are regarded as being fully occupied and can serve as the reference for 100% occupancy. The definition of 0% oxygen occupancy is challenging due to the presence of residual contrast at the oxygen sites. Assuming that the minimum observed phase contrast corresponds to 0% oxygen occupancy, our estimation in Figure R7d suggests that residual oxygen occupancy is mostly in the range of 10–15%. Consequently, the provision of a range of residual oxygen occupancy is based on both simulations and experimental results. The EELS data in Figure 4 also indicates inhomogeneous hole doping, which is associated with the variable residual oxygen content. Figure R7 has been included as supplementary information, accompanied by the relevant discussion, in the paragraph of Figure 3 in the manuscript.

5. In Figure 2, and figure 3, what is the reason that all the depth images are ended with 10 nm in thickness?

Response: We thank the reviewer for this valuable question. The sample thickness was estimated from the CBED patterns, and is approximately 10 ~ 13 nm, as shown in Figure R12. The reconstruction of the phase contrast images in Figures 2 and 3 was performed using a thickness of 10 nm for the sample, which was found to be successful across different slices, as demonstrated in Figure R3.

Figure R12. (a) Experimental convergent beam electron diffraction (CBED) pattern. Simulated CBED patterns for the sample thickness of (b) 10 nm and (c) 13 nm.

6. This sentence does not make sense: In contrast, the apical oxygen distribution in the SrTiO₃ layer (Figure 3d) is fully occupied, whereas the inner NdNiO₂ layer reveals low level residual apical oxygen.

Response: We thank the reviewer for the rigorous and careful comment. We have revised the sentence in the manuscript to improve clarity.

Reviewer #4 (Remarks to the Author):

Response: We thank the co-reviewer for the related insightful comments and suggestions above.

This work reported distorted basal O and apical O distribution along the projection direction using multi-slice ptychography. EELS is complemented to evidence the hole doping of those apical O. I have some questions on the current conclusions and suggestions on developing this paper.

1, Figure 1 is purely on simulated data, this means there is a theoretical model first, then images are simulated based on this. Reading the paragraph of figure 1 indicates a reverse way of expression. I believe the purpose of figure 1 is to demonstrate the apical O with 35% occupancy can be imaged in this model. This model is not clearly described. It seems apical O concentrates at the bottom region. Figure 1a does not show an over-defocus probe.

2, Figure S2 shows very abnormal reconstruction at each slice. The question is why the top slices look like not convergent, although there is a defocus.

3, The authors claim: The discrepancy observed between the phase contrast and the structural model is closely related to sample thickness, and the scattering angles and defocus conditions of the reconstruction, amongst other factors. I do not think they are suitable here, the key reason should be due to the contrast transfer function of the phase methods and the capabilities of coping with the dynamic scattering effects.

4, From figure 2, how the authors conclude the apical O has 12%?

5, In Figure 2, and figure 3, what is the reason that all the depth images are ended with 10 nm in thickness?

6, This sentence does not make sense: In contrast, the apical oxygen distribution in the SrTiO₃ layer (Figure 3d) is fully occupied, whereas the inner NdNiO₂ layer reveals low level residual apical oxygen.